# Towards Fine-grained Molecular Graph-Text Pre-training

## Abstract

Understanding molecular structure and related knowledge is crucial for scientific research. Recent studies integrate molecular graphs with their textual descriptions to enhance molecular representation learning. However, they focus on the whole molecular graph and neglect frequently occurring subgraphs, known as motifs,which are essential for determining molecular properties. Without such fine-grained knowledge, these models struggle to generalize to unseen molecules and tasks that require motif-level insights. To bridge this gap, we propose FineMolTex, a novel **Fine**-grained **Mol**ecular graph-**Tex**t pre-training framework to jointly learn coarse-grained molecule-level knowledge and fine-grained motif-level knowledge. Specifically, FineMolTex consists of two pre-training tasks: a contrastive alignment task for coarse-grained matching and a masked multi-modal modeling task for fine-grained matching. In particular, the latter predicts the labels of masked motifs and words, leveraging insights from each other, thereby enabling FineMolTex to understand the fine-grained matching between motifs and words. Finally, we conduct extensive experiments across three downstream tasks, achieving up to 230% improvement in the text-based molecule editing task. Additionally, our case studies reveal that FineMolTex successfully captures fine-grained knowledge, potentially offering valuable insights for drug discovery and catalyst design.

## 1 Introduction

Comprehending molecular structure and related knowledge is pivotal in scientific investigations spanning diverse fields, including chemistry, drug discovery, and materials science (Gilmer et al., 2017). Recent advancements in artificial intelligence and machine learning have yielded promising outcomes for molecule-based tasks such as retrosynthesis (Yan et al., 2020) and drug discovery (Gilmer et al., 2017). The majority of these studies (Krenn et al., 2020; Duvenaud et al., 2015; Liu et al., 2019a; Toshev et al., 2023; Atz et al., 2021) concentrate solely on the molecular structure, such as SMILES strings, molecular graphs, and geometric structures. They learn molecular representations under supervised signals such as toxicity level and drug activity. However, this supervised learning requires extensive and costly labeling of pre-defined categories, limiting the application of previous methods to unseen categories and tasks.

Fortunately, compared to task-specific labeled data, textual descriptions of molecules are fairly abundant. These descriptions can be found in chemical database annotations, research papers in chemistry and biology, and drug instruction sheets (Liu et al., 2023a), providing general information on molecular usage, efficacy, chemical properties, and even detailed insights into specific functional groups and chemical moieties (Kim et al., 2021). Hence, several studies explore molecular structures along with their corresponding descriptions. MoleculeSTM (Liu et al., 2023a) and MoMu (Su et al., 2022) align the whole molecular graphs with their textual descriptions employing a contrastive learning approach as shown in Figure 1(a). MolCA (Liu et al., 2023b) further utilizes a cross-modal projector to map the graph embedding space to the input space of the language model. In this way, these studies reduce the reliance on task-specific labels.

However, these approaches primarily focus on the overall structure of the molecule level, failing to capture fine-grained knowledge of the sub-molecule level, such as functional groups. A natural tool to model sub-molecular structures is the motif (Zhang et al., 2021), which refers to frequently recurring, significant subgraphs within molecular graphs. Motifs often play a key role in determining the properties of the whole molecular graph (Zhang et al., 2021), and motif-level knowledge is frequently

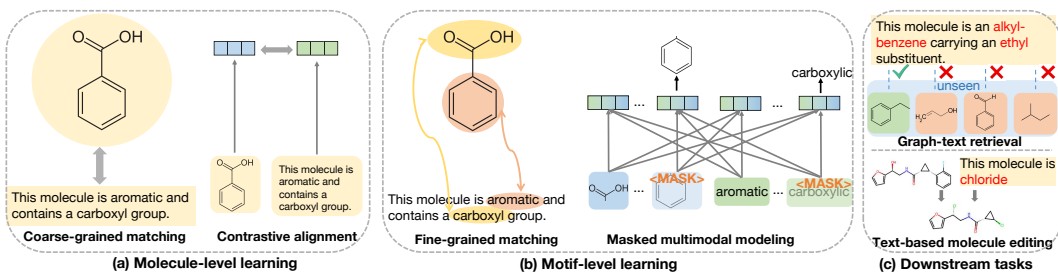

Figure 1: Comparison of molecule- and motif-level learning, and illustration of downstream tasks.

depicted in textual descriptions. As shown in Figure 1(b), a benzene ring indicating aromaticity property is reflected by the mention of "aromatic", and a carboxyl group is reflected by its name "carboxyl" in the description, revealing a fine-grained matching between motifs and texts.

Modeling the fine-grained motif-level knowledge is crucial for two reasons. First, motif-level knowledge is necessary for the generalization to unseen molecules, which are still largely composed of various motifs that have been seen before. For example, consider the zero-shot graph-text retrieval task shown in Figure 1(c), which aims to find the molecule most relevant to the given text. Even if the model has not been trained on the candidate molecules, it has seen many of the motifs within the unseen molecules such as the benzene and the ethyl group, corresponding to the words "benzene" and "carboxylic", respectively. Thus, the model can easily recognize the relevant molecule. Second, it bridges the gap for downstream tasks that require fine-grained knowledge. For example, in the molecule editing task illustrated in Figure 1(c), the model aims to modify part of the molecular structure based on textual instruction. This requires the model to understand the names or properties of the motifs like "chloride".

Despite the significance of this fine-grained knowledge, it is challenging to jointly learn both molecule- and motif-level knowledge, and also non-trivial to capture fine-grained matching without supervised signals. To overcome these issues, in this work, we propose a novel **Fine**-grained **Mol**ecular graph-**Tex**t framework (**FineMolTex**) to learn fine-grained motif-level knowledge, as well as coarse-grained molecule-level knowledge. First, we extract each motif or word token with an individual embedding for fine-grained knowledge and utilize two global tokens to holistically represent a molecular graph and its corresponding text description as coarse-grained knowledge. Second, to align coarse- and fine-grained matching, we propose two pre-training tasks respectively: the contrastive alignment task based on the global tokens, and the masked multi-modal learning task based on the motif and word tokens. Specifically, in the masked multi-modal modeling task illustrated in Figure 1(b), we randomly mask some motif and word tokens, and further incorporate a cross-attention transformer layer to integrate the embeddings of motifs and words. By predicting the labels of masked motifs and words based on information from each other, the learning of fine-grained alignment knowledge is enhanced. In summary, we outline our contributions as follows:

- We are the first to reveal that learning fine-grained motif-level knowledge provides key insight for bridging molecular graphs and text descriptions, further empowering the ability to generalize to unseen molecules and tasks.

- We introduce a novel framework named FineMolTex, consisting of two self-supervised pre-training tasks, to simultaneously learn coarse- and fine-grained knowledge. In particular, the masked multi-modal learning task enhances the prediction for masked tokens leveraging information from the other modality, promoting the learning of fine-grained alignment information.

- Experimental results across three downstream tasks underscore the effectiveness of FineMolTex, with a notable improvement of up to 230% in the text-based molecule editing task. Furthermore, case studies demonstrate that FineMolTex effectively aligns motifs and words, further facilitating applications such as drug discovery and catalyst design.

## 2 RELATED WORK

We provide a brief review on molecular multi-modal learning. Prior works predominantly concentrate on modeling the chemical structures such as 1D SMILES (Krenn et al., 2020), 2D molecular graphs

(Duvenaud et al., 2015; Liu et al., 2019a; Zhang et al., 2021), and 3D geometric structures (Toshev et al., 2023; Atz et al., 2021; Wang et al., 2023). They utilize supervised signals on a predetermined set, and thus cannot generalize to unseen categories without labeled examples. Recently, KV-PLM (Zeng et al., 2022) bridges this gap by linking SMILES with biomedical texts through a unified language modeling framework. Nonetheless, 1D SMILES may omit certain structural details and fail to identify structural similarities among molecules due to its non-uniqueness. To address these limitations, MoleculeSTM (Liu et al., 2023a) and MoMu (Su et al., 2022) employ a contrastive learning approach to align the molecular graph with its corresponding text, thus performing well on unseen molecules and texts. However, these models are less effective on molecule-to-text generation tasks because language models are not yet well-versed in interpreting graphs as generative conditions. Therefore, MolCA (Liu et al., 2023b) introduces a cross-modal projector to align the embedding space of the molecular graph with the language model's input space, enabling the comprehension of 2D graphs as generative conditions. This approach has also been extended to 3D graph structures, where 3D-MoLM (Li et al., 2024) uses a cross-modal projector to synchronize the embedding space of the 3D geometric structure with that of the language model. Additionally, various efforts have been devoted to tackling specific molecular tasks based on textual data, including zero-shot instruction molecular learning (Zhao et al., 2023), molecular reaction prediction (Shi et al., 2023), and molecular relational modeling (Fang et al., 2024).

More related works on graph-based molecular learning, as well as more general multi-modal learning, can be found in Appendix A.

## 3 THE PROPOSED APPROACH

We propose FineMolTex, a novel fine-grained molecular graph-text framework, learning both molecule- and motif-level knowledge. The model architecture is outlined in Figure 2. This section first introduces the key components in the architecture and then describes the two pre-training tasks.

### 3.1 KEY COMPONENTS OF FINEMOLTEX

To capture coarse- and fine-grained knowledge, we propose FineMolTex, consisting of five key components: 1) the tokenization component to decompose molecular graphs and texts into motif and word tokens; 2) a graph encoder to capture the structure of molecules and motifs; 3) a text encoder to extract the knowledge from texts and words, 4) a cross-attention layer to integrate information from different modalities; 5) a Transformer layer to generate embeddings for each token based on its contextual tokens from the same modality.

**Tokenization.** As shown in Figure 2, for fine-grained modeling, we fragment the molecular graphs and texts into motif tokens and word tokens. We employ the BRICS (Degen et al., 2008) algorithm to transform the molecular graph into a motif tree, and then generate a motif sequence following a breadth-first search order. Then we utilize the post-processing procedure (Zhang et al., 2021) to consolidate the motif vocabulary. We break the textual description into word tokens using the word tokenizer of SciBERT (Beltagy et al., 2019). For coarse-grained modeling, the global tokens of molecule and text, <MOL> and <CLS>, are inserted at the beginning of the motif and word sequences, respectively, resulting in the sequences $m_0, m_1, \ldots, m_J$ and $t_0, t_1, \ldots, t_D$, where $J$ and $D$ are the lengths of the sequences.

**Graph Encoder.** Let $\mathcal{G} = (\mathcal{V}, \mathcal{E}, \mathbf{X})$ represent a molecular graph with $N$ atoms, where $\mathcal{V} = \{v_1, v_2, \ldots, v_N\}$ is the set of atoms, $\mathcal{E} \subseteq \mathcal{V} \times \mathcal{V}$ denotes the bonds, and $\mathbf{X} = [\mathbf{x_1}, \mathbf{x_2}, \ldots, \mathbf{x_N}] \in \mathbb{R}^{N \times \zeta}$ is the atom feature matrix. Here, $\mathbf{x_i}$ is the feature vector of atom $v_i$, and $\zeta$ is the dimension of atom features. We utilize GraphMVP (Liu et al., 2019b), a pre-trained Graph Isomorphism Network (GIN), to encode each motif token. GraphMVP employs multi-view pre-training to connect 2D topologies and 3D geometries, leveraging the GEOM dataset (Axelrod & Gómez-Bombarelli, 2022), which contains 250K molecular conformations. Denoting the GraphMVP encoder as $f_{\text{GraphMVP}}$, we encode each atom $v$ into an embedding as follows:

$$\mathbf{g_v} = f_{\text{GraphMVP}}(\mathbf{x_v}, \mathbf{x_u}), \ u \in \mathcal{N}(v), \tag{1}$$

where $\mathcal{N}(v)$ denotes the set of neighboring atoms of $v$. Then we pool the atom embeddings into a motif-level embedding, $\mathbf{g}_{\mathcal{G}}$, as follows:

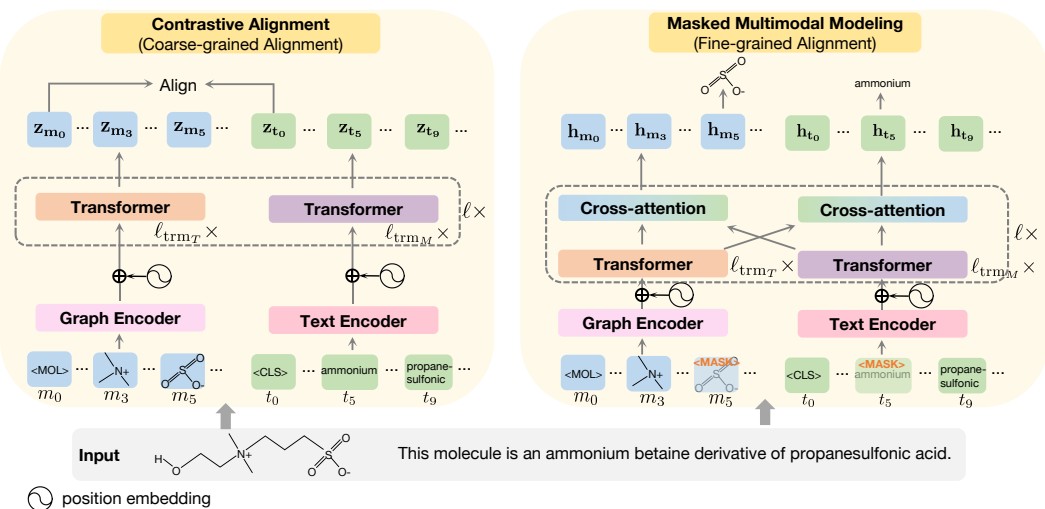

Figure 2: Architecture of FineMolTex. The input is a graph-text pair with both a molecular structure and a corresponding description. The components in the same color share the same weights.

$$\mathbf{g}_{\mathcal{G}} = \text{READOUT}(\{\mathbf{g_v}|v \in \mathcal{G}\}), \quad \text{for} \quad \mathcal{G} \in \{m_1, m_2, \ldots, m_J\}, \tag{2}$$

where $\text{READOUT}(\cdot)$ is permutation invariant, implemented as the average function in our model.

To preserve the intrinsic connectivity of the motifs in the original molecule, we generate position embeddings based on the breadth-first search order and incorporate them into the motif embeddings $\mathbf{g_{m_0}}, \mathbf{g_{m_1}}, \ldots, \mathbf{g_{m_J}}$, resulting in updated embeddings $\mathbf{g'_{m_0}}, \mathbf{g'_{m_1}}, \ldots, \mathbf{g'_{m_J}}$.

**Text Encoder.** We use SciBERT (Beltagy et al., 2019), which has been pre-trained on texts from the chemical and biological domains, as our text encoder, denoted as $f_{\text{bert}}$. It can encode a text sequence as:

$$\mathbf{b_{t_0}}, \mathbf{b_{t_1}}, \ldots, \mathbf{b_{t_D}} = f_{\text{bert}}(t_0, t_1, \ldots, t_D). \tag{3}$$

Subsequently, we add the position embeddings to the token embeddings following previous work (Beltagy et al., 2019), yielding $\mathbf{b'_{t_0}}, \mathbf{b'_{t_1}}, \ldots, \mathbf{b'_{t_D}}$.

**Transformer Layer.** To capture the contextual information for each token, we use "encoder-style" Transformer layers (Vaswani et al., 2017), which consist of a multi-head self-attention layer and a fully connected feed-forward network. This enables the tokens to gather information from other tokens in the same modality. We utilize $f_{\text{trm}_T}$ and $f_{\text{trm}_M}$ for the text and molecule modality, respectively, as follows.

$$\mathbf{z_{t_0}}, \mathbf{z_{t_1}}, \ldots, \mathbf{z_{t_D}} = f_{\text{trm}_T}(\mathbf{b'_{t_0}}, \mathbf{b'_{t_1}}, \ldots, \mathbf{b'_{t_D}}), \tag{4}$$

$$\mathbf{z_{m_0}}, \mathbf{z_{m_1}}, \ldots, \mathbf{z_{m_J}} = f_{\text{trm}_M}(\mathbf{g'_{m_0}}, \mathbf{g'_{m_1}}, \ldots, \mathbf{g'_{m_J}}). \tag{5}$$

**Cross-attention Layer.** We integrate information from different modalities via cross-attention layers $f_{\text{crs}_M}$ and $f_{\text{crs}_T}$ for molecular graph and text, respectively. Consider the cross-attention layer $f_{\text{crs}_M}$ for molecular graph: the queries are from the same modality, $Q_m = Z_m W_m^Q$, while the keys and values are from the text modality, $K_t = Z_t W_t^K$ and $V_t = Z_t W_t^V$. Here $W_m^Q, W_t^K, W_t^V$ are learnable weights, $Z_m = [\mathbf{z_{m_0}}, \mathbf{z_{m_1}}, \ldots, \mathbf{z_{m_J}}]$, and $Z_t = [\mathbf{z_{t_0}}, \mathbf{z_{t_1}}, \ldots, \mathbf{z_{t_K}}]$. Subsequently, the output of scaled dot-product attention is computed as:

$$\text{Attention}(Q_m, K_t, V_t) = \text{softmax}\left(\frac{Q_m K_t^T}{\sqrt{d_k}}\right) V_t, \tag{6}$$

where $d_k$ is the dimension of queries and keys. The cross-attention layer for text is designed similarly. Hence, the encoding of each token accounts for tokens from the other modality, enabling the learning of fine-grained alignment at the motif level. The outputs of the cross-attention layer are:

$$\mathbf{h_{t_0}}, \mathbf{h_{t_1}}, \ldots, \mathbf{h_{t_D}} = f_{\text{crs}_T}(\mathbf{z_{t_0}}, \mathbf{z_{t_1}}, \ldots, \mathbf{z_{t_D}}), \tag{7}$$

$$\mathbf{h_{m_0}}, \mathbf{h_{m_1}}, \ldots, \mathbf{h_{m_J}} = f_{\text{crs}_M}(\mathbf{z_{m_0}}, \mathbf{z_{m_1}}, \ldots, \mathbf{z_{m_J}}). \tag{8}$$

## 3.2 PRE-TRAINING TASKS

We propose two pre-training tasks, the contrastive alignment task for coarse-grained alignment, and the masked multi-modal modeling task for fine-grained alignment.

**Contrastive Alignment.** For coarse-grained alignment at the molecule level, we align the graph-text pairs from the same molecules and contrast the pairs from different molecules, which can be achieved by optimizing the following loss:

$$
L_{\text{con}} = -\frac{1}{2}\mathbb{E}_{m_0,t_0}\left[\log\frac{\exp(\cos(\mathbf{z}_{\mathbf{m_0}},\mathbf{z}_{\mathbf{t_0}})/\tau)}{\exp(\cos(\mathbf{z}_{\mathbf{m_0}},\mathbf{z}_{\mathbf{t_0}})/\tau)+\sum_{t_0'}\exp(\cos(\mathbf{z}_{\mathbf{m_0}},\mathbf{z}_{\mathbf{t_0'}})/\tau)}\right.
$$
$$
\left.+\log\frac{\exp(\cos(\mathbf{z}_{\mathbf{t_0}},\mathbf{z}_{\mathbf{m_0}})/\tau)}{\exp(\cos(\mathbf{z}_{\mathbf{t_0}},\mathbf{z}_{\mathbf{m_0}})/\tau)+\sum_{m_0'}\exp(\cos(\mathbf{z}_{\mathbf{t_0}},\mathbf{z}_{\mathbf{m_0'}})/\tau)}\right], \tag{9}
$$

where $\mathbf{z}_{\mathbf{m_0}}$, $\mathbf{z}_{\mathbf{m_0'}}$, $\mathbf{z}_{\mathbf{t_0}}$, and $\mathbf{z}_{\mathbf{t_0'}}$ denote the output embeddings from the Transformer layer, $t_0'$ and $m_0'$ are the negative instances sampled from the same batch of graph-text pairs, and $\cos(\cdot,\cdot)/\tau$ is the cosine similarity scaled by the temperature hyperparameter $\tau$. In this way, we capture the molecule-level knowledge, aligning the embedding space of molecular graphs and texts holistically.

**Masked Multi-modal Modeling.** For fine-grained alignment at the motif level, we mask some of the tokens and predict their labels. Based on the fragmented motifs of all molecules in the pre-training dataset, we construct a motif dictionary that includes motifs along with their labels and frequency. Then we randomly mask approximately 20% of the motif tokens that have neither too high nor too low a frequency in the motif dictionary, as well as 15% of the word tokens. The token embeddings of the motifs and words are updated utilizing $\ell_{\text{trm}_M}$ and $\ell_{\text{trm}_T}$ transformer layers, respectively. Subsequently, information from the two modalities is integrated via our cross-attention layer. This entire process is iterated for $\ell$ times.

Based on the output embeddings of fine-grained tokens from the cross-attention layer $\mathbf{h}_{\mathbf{t_1}},\ldots,\mathbf{h}_{\mathbf{t_D}}$ and $\mathbf{h}_{\mathbf{m_0}},\mathbf{h}_{\mathbf{m_1}},\ldots,\mathbf{h}_{\mathbf{m_J}}$, we utilize two classifiers $\rho_m$ and $\rho_t$ to predict the labels of the masked motifs and words: $\hat{y}_{m_i} = \rho_m(\mathbf{h}_{\mathbf{m_i}})$, $\hat{y}_{t_j} = \rho_t(\mathbf{h}_{\mathbf{t_j}})$, where $\hat{y}_{m_i}$ is the predicted label of motif $m_i$, and $\hat{y}_{t_j}$ is the predicted label of word $t_j$. Given the ground truth labels $y_{m_i}$ and $y_{t_j}$, the model is trained by reconstructing the masked tokens as:

$$
L_{\text{pre}} = \beta\sum_i \text{CE}(\hat{y}_{m_i},y_{m_i}) + \alpha\sum_j \text{CE}(\hat{y}_{t_j},y_{t_j}), \tag{10}
$$

where $\alpha$, $\beta$ are hyperparameters, and $\text{CE}(\cdot,\cdot)$ is the cross-entropy loss. The key to achieving fine-grained alignment lies in the cross-attention layer, which enables the model to predict the labels of masked tokens based on tokens from the other modality. For instance, as illustrated in Figure 2, predicting the label of $SO_3^-$ solely based on the unmasked motif tokens is challenging. However, by leveraging the embeddings of word tokens, particularly "propanesulfonic" which includes the $SO_3^-$ group, we can gain relevant information about the masked token. Consequently, the model implicitly learns fine-grained alignment knowledge, thereby augmenting its motif-level knowledge.

**Overall Loss.** FinMolTex is optimized by the overall loss $L = L_{\text{con}} + L_{\text{pre}}$. Thus, FineMolTex is able to jointly learn the molecule- and motif-level knowledge.

## 4 EXPERIMENTS

In this section, we conduct extensive experiments to demonstrate the effectiveness of FineMolTex. Before evaluating, we first conduct the two pre-training tasks on the PubChemSTM dataset (Liu et al., 2023a), which includes 281K graph-text pairs from PubChem (Kim et al., 2021). Each molecular graph is paired with a textual description that elaborates on its chemical and physical properties or highlights its high-level bioactivities. Details of the pre-training data and process can be found in Appendix C.1.1 and C.4.

The goal of our experiments is to answer the following research questions (RQs).

**RQ1.** Can FineMolTex better generalize to unseen molecules?
**RQ2.** Can FineMolTex bridge the gap to tasks centered on motif-level knowledge?

Table 1: Accuracy (%$\pm\sigma$) of graph-text retrieval task on DrugBank-Pharmacodynamics.

| | Given Molecular Graph | | | Given Text | | |
|---|---|---|---|---|---|---|
| $T$ | 4 | 10 | 20 | 4 | 10 | 20 |
| KV-PLM | 68.38±0.03 | 47.59±0.03 | 36.54±0.03 | 67.68±0.03 | 48.00±0.02 | 34.66±0.02 |
| MolCA | 83.75±0.54 | 74.25±0.26 | 66.14±0.21 | 81.27±0.33 | 69.46±0.17 | 62.13±0.16 |
| MoMu-S | 70.51 ±0.04 | 55.20±0.15 | 43.78±0.10 | 70.71±0.22 | 54.70±0.31 | 44.25±0.43 |
| MoMu-K | 69.40 ±0.11 | 53.14±0.26 | 42.32±0.28 | 68.71±0.03 | 53.29±0.05 | 43.83±0.12 |
| MoleculeSTM | 92.14±0.02 | 86.27±0.02 | 81.08±0.05 | 91.44±0.02 | 86.76±0.03 | 81.68±0.05 |
| FineMolTex | **95.86±0.34** | **91.95±0.06** | **85.80±0.05** | **95.80±0.06** | **92.18±0.12** | **85.01±0.32** |

Table 2: Accuracy (%$\pm\sigma$) of graph-text retrieval task on molecule-ATC.

| | Given Molecular Graph | | | Given Text | | |
|---|---|---|---|---|---|---|
| $T$ | 4 | 10 | 20 | 4 | 10 | 20 |
| KV-PLM | 60.94±0.00 | 42.35±0.00 | 30.32±0.00 | 60.67 ±0.00 | 40.19±0.00 | 29.02±0.00 |
| MolCA | 67.34±0.05 | 53.51±0.12 | 44.10±0.03 | 65.18±0.34 | 51.01±0.26 | 41.30±0.51 |
| MoMu-S | 64.72±0.04 | 48.72±0.03 | 37.64±0.02 | 64.98±0.13 | 49.58±0.05 | 39.04±0.16 |
| MoMu-K | 61.79±0.14 | 45.69±0.22 | 34.55±0.09 | 63.32±0.15 | 47.55±0.06 | 37.68±0.18 |
| MoleculeSTM | 69.33±0.03 | 54.83±0.04 | 44.13±0.05 | 71.81±0.05 | 58.34±0.07 | 47.58±0.05 |
| FineMolTex | **75.43±0.15** | **60.66±0.08** | **49.45±0.24** | **75.22±0.12** | **60.29±0.04** | **48.42±0.15** |

**RQ3.** Can FineMolTex perform better on single-modality tasks?
**RQ4.** Has FineMolTex learned fine-grained knowledge?
**RQ5.** Are the token masking and cross-attention layers beneficial?

## 4.1 GENERALIZATION TO UNSEEN MOLECULES (RQ1)

To answer RQ1, we conduct a **zero-shot graph-text retrieval task** to examine the generalizability of FineMolTex on unseen molecules and texts. Given a molecular graph and $T$ candidate textual descriptions, the goal is to identify the textual description that best aligns with the molecular graph. Conversely, given a textual description and $T$ candidate molecular graphs, identify the molecular graph that best matches the text. This task can be addressed by calculating the similarity of the molecular graphs and texts in the joint embedding space, thus allowing zero-shot inference.

**Datasets and Baselines.** We utilize DrugBank-Pharmacodynamics, molecule-ATC, and DrugBank-Description (Liu et al., 2023a) extracted from the DrugBank database (Wishart et al., 2018) for evaluation. These datasets include molecular graphs and their chemical descriptions. Details of the datasets can be found in Appendix C.1.2. We compare with five multimodal molecular models: KV-PLM (Zeng et al., 2022), MolCA (Liu et al., 2023b), MoMu-S (Su et al., 2022), MoMu-K (Su et al., 2022), and MoleculeSTM (Liu et al., 2023a). Specifically, KV-PLM uses SMILES to represent the structure of the molecule, while others use graph structures.

**Results.** We report the results on the first two datasets in Tables 1 and 2, and defer those on DrugBank-Description to Appendix D.1 due to space limit. We make the following observations. 1) Across different values of $T$, FineMolTex consistently outperforms the baselines that neglect motif-level knowledge. The superior performance demonstrates that fine-grained motif-level knowledge facilitates generalization to unseen molecules, which likely contain seen motifs. 2) FineMolTex maintains strong performance in both directions (given graph, and given text). The symmetry further indicates that the embedding spaces of both modalities are well-aligned and similarly well-learned. 3) We observe that KV-PLM, which utilizes SMILES to capture molecular structures, is less effective than other models employing graphs, consistent with previous findings (Liu et al., 2023a) that 2D graph structure is more expressive than 1D SMILES.

## 4.2 APPLICATION TO MOTIF-CENTERED TASKS (RQ2)

To answer RQ2, we employ a zero-shot **text-based molecule editing task**, which is highly relevant to practical applications including catalyst design and targeted drug discovery. Specifically, we utilize FineMolTex to collaborate with a molecule generation module, following the design in (Liu et al.,

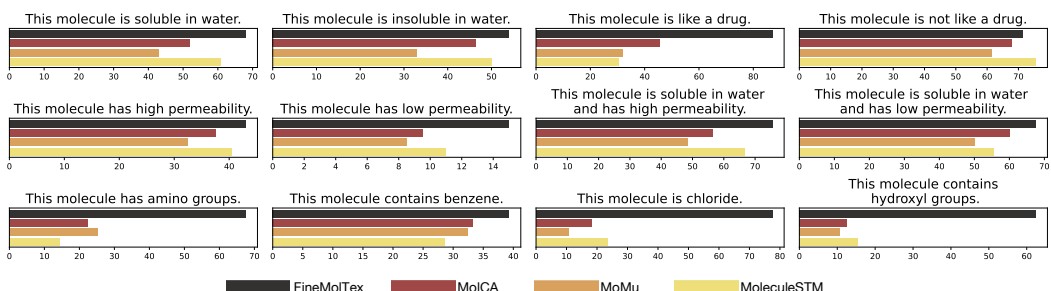

Figure 3: Hit ratios of 12 text-based molecule editing tasks.

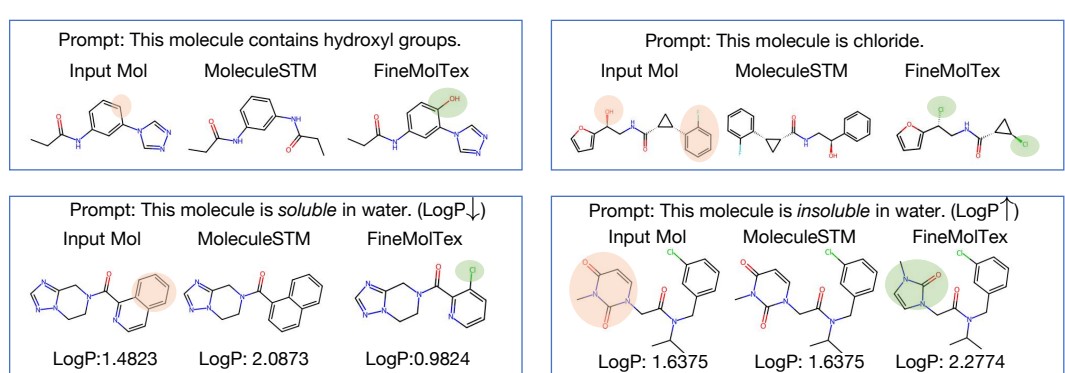

Figure 4: Visual analysis of the output molecules of MoleculeSTM and Motif-MolTex on 4 text-based molecule editing tasks. Differences between the input and output molecule of FineMolTex are highlighted in red and green circles. Lower LogP indicates higher water solubility.

2023a), to modify a specified molecule according to a text prompt. Hence, motif-level knowledge is essential for this task, as the model needs to replace certain motifs with others that are related to specific properties and names as indicated in the text prompt. We defer the technical details to Appendix B. We randomly sample 200 molecules from ZINC (Irwin et al., 2012), and select 12 text prompts, including 8 prompts pertaining to physical properties (Liu et al., 2023a), and 4 based on the names of the motifs. We utilize MoleculeSTM, MoMu, and MolCA as baselines.

**Evaluation.** We employ different methods to assess whether the generated molecules satisfy the two types of prompts. For the 8 prompts on physical properties, we employ three measures: LogP, QED, and tPSA, which measures solubility (Leo et al., 1971), drug-likeness (Bickerton et al., 2012), and permeability (Ertl et al., 2000), respectively. We consider the editing to be successful if the difference in measurements between the input and output molecules exceeds a specified threshold $\Delta$, which we have set to 0 following one of the settings in literature (Liu et al., 2023a). For the 4 prompts based on motif names, we use RDKit (Landrum, 2024) to verify the presence of the indicated motifs. For all 12 prompts, we report the *hit ratio*: the proportion of generated molecules that meet our expectations.

**Results.** As shown in Figure 3, FineMolTex shows superior performance on these prompts, especially on the 4 prompts with motif names. Notably, we achieve a relative gain of up to 230% over the best-performing baseline, demonstrating that FineMolTex has an advanced understanding of motif-level knowledge. We also visualize the output molecules of MoleculeSTM and FineMolTex in Figure 4. It can be observed that while MoleculeSTM produces incorrect molecules, FineMolTex accurately generates the intended molecules. Specifically, when prompted to generate molecules that are soluble in water, FineMolTex successfully creates molecules with lower LogP than the input molecule, as lower LogP indicates higher water solubility. Similarly, when prompted to generate molecules with hydroxyl groups or chlorine atoms, FineMolTex correctly does so. These results confirm that FineMolTex possesses a deeper understanding of motif-level knowledge, thereby enhancing the generative capabilities. More visual results can be seen in Appendix D.2.

Table 3: Downstream results ($\%\pm\sigma$) on eight binary classification datasets from MoleculeNet.

| Model | BBBP | Tox21 | ToxCast | Sider | ClinTox | MUV | HIV | Bace | Avg |
|-------|------|-------|---------|-------|---------|-----|-----|------|-----|
| AttrMask | 67.8±2.6 | 75.0±0.2 | 63.6±0.8 | 58.1±1.2 | 75.4±8.8 | 73.8±1.2 | 75.4±0.5 | 80.3±0.0 | 71.2 |
| ContextPred | 63.1±3.5 | 74.3±0.2 | 61.6±0.5 | 60.3±0.8 | 80.3±3.8 | 71.4±1.4 | 70.7±3.6 | 78.8±0.4 | 70.1 |
| InfoGraph | 64.8±0.6 | 76.2±0.4 | 62.7±0.7 | 59.1±0.6 | 76.5±7.8 | 73.0±3.6 | 70.2±2.4 | 77.6±2.0 | 70.0 |
| MolCLR | 67.8±0.5 | 67.8±0.5 | 64.6±0.1 | 58.7±0.1 | 84.2±1.5 | 72.8±0.7 | 75.9±0.2 | 71.1±1.2 | 71.3 |
| GraphMVP | 68.1±1.4 | 77.1±0.4 | 65.1±0.3 | 60.6±0.1 | 84.7±3.1 | 74.4±2.0 | 77.7±2.5 | 80.5±2.7 | 73.5 |
| GraphCL | 69.7±0.7 | 73.9±0.7 | 62.4±0.6 | 60.5±0.9 | 76.0±2.7 | 69.8±2.7 | 78.5±1.2 | 75.4±1.4 | 70.8 |
| KV-PLM | 70.5±0.5 | 72.1±1.0 | 55.0±1.7 | 59.8±0.6 | 89.2±2.7 | 54.6±4.8 | 65.4±1.7 | 78.5±2.7 | 68.2 |
| MoMu-S | 70.5±2.0 | 75.6±0.3 | 75.6±0.3 | 60.5±0.9 | 79.9±4.1 | 70.5±1.4 | 75.9±0.8 | 76.7±2.1 | 71.6 |
| MoMu-K | 70.1±1.4 | 75.6±0.5 | 63.0±0.4 | 60.4±0.8 | 77.4±4.1 | 71.1±2.7 | 76.2±0.9 | 77.1±1.4 | 71.4 |
| MolCA | 70.0±0.5 | 77.2±0.5 | 64.5±0.8 | 63.0±1.7 | 89.5±0.7 | 72.1±1.3 | 77.2±0.6 | 79.8±0.5 | 74.2 |
| MoleculeSTM | 70.0±0.5 | 76.9±0.5 | 65.1±0.4 | 61.0±1.05 | 92.5±1.1 | 73.4±2.9 | 77.0±1.8 | 80.8±1.3 | 74.6 |
| FineMolTex | **71.0±0.4** | **77.1±0.5** | **66.0±1.2** | **65.0±2.3** | **92.7±0.8** | **76.2±1.2** | **78.9±0.6** | **84.6±1.4** | **76.4** |

## 4.3 APPLICATION TO SINGLE-MODALITY TASK (RQ3)

While FineMolTex can simultaneously utilize pre-trained knowledge from both graphs and texts, we also verify its effectiveness on single-modality tasks, namely, **molecular property prediction tasks**. We use MoleculeNet (Wu et al., 2018) as the dataset, which only provides molecular graphs as input without texts. More specifically, there are eight binary classification tasks, and we report ROC-AUC for evaluation. More detailed dataset descriptions are provided in Appendix C.1.3.

**Baselines.** We compare FineMolTex against nine baselines, including 1) five pre-trained GNN models: AttrMasking (Hu et al., 2019), ContextPred (Hu et al., 2019), InfoGraph (Sun et al., 2019), MolCLR (Wang et al., 2021b), and GraphMVP (Liu et al., 2019b); 2) three graph-text multimodal models: MoMu-S (Su et al., 2022), MoMu-K (Su et al., 2022), and MoleculeSTM (Liu et al., 2023a), and 3) one SMILES-text multimodal model: KV-PLM (Zeng et al., 2022).

**Results.** As shown in Table 3, FineMolTex consistently outperforms all baselines, achieving relative gains of 3.2%, 2.4%, and 4.7% on SIDER, MUV, and BACE, respectively, compared to the best baseline. The promising performance of FineMolTex indicates that it implicitly utilizes pre-trained knowledge from the text modality even when the input consists solely of graphs. Additionally, KV-PLM exhibits a notable performance gap from other models, due to its use of 1D SMILES strings for molecular structure and a smaller pre-training dataset.

## 4.4 ANALYSIS OF LEARNED FINE-GRAINED KNOWLEDGE (RQ4)

We evaluate whether FineMolTex captures fine-grained alignment information in the joint embedding space, and assess if it can predict the labels of masked motifs based on fine-grained knowledge.

**Visualization of Motif and Word Embeddings.** To evaluate whether FineMolTex captures fine-grained alignment knowledge, we select motif and word tokens from 1,500 graph-text pairs in the PubChemSTM dataset, excluding meaningless words such as "this" and "a". In total, we visualize 3,469 motif tokens and 6,914 text tokens with $t$-SNE (Maaten & Hinton, 2008) in Figure 5a, where triangles denote text tokens, and circles denote motif tokens, with different colors indicating various labels. To examine the details of the tokens, we zoom into several regions in the figure, retaining only the colors and legends of the tokens we are interested in. For brevity, we utilize SMILES to represent the motif structures. We observe that text and motif tokens corresponding to each other are also close in the embedding space. For instance, in the pink frame, the word "ammonium" is close to the motif tokens "[NH2+]=O", "C=[NH2+]", and "[NH3+]O", which are related to "ammonium." In the blue frame, the word "poison" is adjacent to the motifs "[AsH3]" and "O[AsH2]", which are poisonous. In the orange frame, the word "sulf" is close to the motif tokens "OS", "CCSSCC", and "CC1=CSC(C)=N1", all of which represent sulfides. These results demonstrate that FineMolTex learns the connections between motifs and their chemical names and properties, thereby significantly enhancing its expressiveness.

**Predictions Based on Fine-grained knowledge.** To further verify that FineMolTex can utilize the learned fine-grained alignment knowledge for predictions, we utilize Local Interpretable Model-Agnostic Explanation (LIME) (Ribeiro et al., 2016), a well-established tool that can explain the

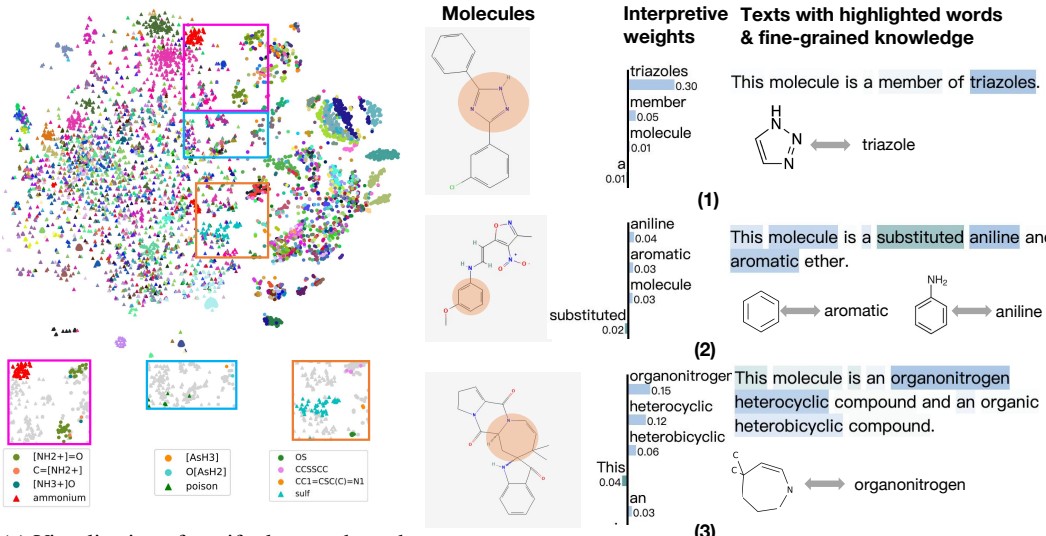

(a) Visualization of motif tokens and word tokens using $t$-SNE. Triangles denote word tokens; circles denote motif tokens.

(b) Explaination of the prediction of certain masked motifs based on text tokens utilizing LIME.

Figure 5: Case studies validating motif-level knowledge learned by our model.

Table 4: Ablation study ($\%\pm\sigma$) on molecule-ATC and DrugBank-Pharmacodynamics.

| | molecule-ATC | | DrugBank-Pharmacodynamics | |
| --- | --- | --- | --- | --- |
| | Given Molecular Graph | Given Text | Given Molecular Graph | Given Text |
| w/o mmm | 70.33±0.28 | 70.86±0.16 | 92.56±0.10 | 92.12±0.25 |
| motif mask only | 72.85±0.46 | 71.15±0.24 | 93.45±0.44 | 92.98±0.18 |
| word mask only | 74.27±0.07 | 72.11±0.13 | 94.86±0.32 | 93.88±0.15 |
| w/o cross-attention | 71.44±0.32 | 70.92±0.18 | 92.66±0.08 | 92.85 ±0.24 |
| FineMolTex | **75.43±0.15** | **75.22±0.12** | **95.86±0.34** | **95.80±0.06** |

predictions of certain masked motifs based on text tokens. By perturbing the input text, LIME observes how the model's predictions change with variations in the input text. Then, LIME fits these perturbed texts and the prediction results to an interpretable model such as a linear model. This approach allows us to quantify the significance of each text token in predicting the motifs, thereby revealing the fine-grained knowledge learned by FineMolTex. The results are shown in Figure 5b (with more cases in Appendix D.4), where text tokens with higher interpretive weights are more crucial for predictions, and thus more relevant to the masked motifs. Specifically, the word with the highest interpretive weights in (1) is "triazoles", which directly is to the name of the masked motif. In (2), the word "aniline" refers to another motif that is very similar to the masked motif, and "aromatic" is the property related to the masked motif. In (3), the masked motif is a kind of "organonitrogen". These findings demonstrate that FineMolTex has effectively acquired motif-level knowledge.

### 4.5 ABLATION STUDY FOR MASKING AND CROSS-ATTENTION LAYERS (RQ5)

To thoroughly explore the impact of the key components in FineMolTex, we compare to several variants, including **w/o mmm**, which drops the masked multimodal modeling task altogether; **motif mask only**, which only mask motif tokens; **word mask only**, which only mask word tokens; **w/o cross-attention**, which excludes cross-attention layers. We evaluate these variants on the graph-text retrieval task used in RQ1, on two datasets with $T = 4$. As reported in Table 4, FineMolTex consistently surpasses the other variants. Specifically, without the masked multimodal modeling task, "w/o mmm" fails to capture fine-grained knowledge at all, resulting in the poorest performance. "motif mask only" and "word mask only" outperform "w/o mmm," because they still enable some level of fine-grained knowledge learning by either predicting motifs based on word tokens or predicting word tokens based on motif tokens. However, they are less effective than FineMolTex, which masks

both words and motifs for mutual alignment. Lastly, without the cross-attention layers, "w/o cross-attention" cannot integrate token embeddings from different modalities, thereby hampering its ability to effectively learn fine-grained knowledge. These observations demonstrate the effectiveness of each component.

Additional experimental results, including further ablation studies and a comparison of pre-training and inference times, are provided in Appendix D.

## 5 CONCLUSIONS

In this paper, we reveal that fine-grained motif-level knowledge is crucial for molecular representation learning. We propose FineMolTex to jointly learn both coarse- and fine-grained knowledge through a contrastive alignment task and a masked multimodal learning task, respectively. By masking the fine-grained tokens and predicting their labels using tokens from the other modality, we can effectively learn fine-grained alignment between motifs and words. Experimental results on three downstream tasks and two case studies demonstrate the effectiveness of FineMolTex.

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
