# OpenReview forum: "Towards Fine-grained Molecular Graph-Text Pre-training"
_ICLR.cc/2025/Conference — ICLR 2025 Conference Withdrawn Submission_

### Official Review · Reviewer_ELaY · 2024-10-30

**Soundness:** 3
**Presentation:** 3
**Contribution:** 2
**Rating:** 5
**Confidence:** 4

**Summary:**

The main idea of this paper is to show that fine-grained motif-level knowledge is crucial for molecular representation learning. Basen on this, the author propose FineMolTex, which jointly learns both coarse and fine-grained knowledge through a contrastive alignment task and a masked multimodal learning task. Experimental results on three downstream tasks and two case studies demonstrate the effectiveness of FineMolTex.

**Strengths:**

1. Unlike previous models that primarily focus on molecule-level representations, the proposed FineMolTex incorporates motif-level knowledge, capturing the significance of frequently occurring subgraphs within molecular graphs. This allows the model to better generalize to unseen molecules, achieving better performance on zero-shot tasks.
2.  This paper is well-written, and each component is clearly presented. The author performed extensive experiment evaluations, and showed that FineMolTex achieved good performance.

**Weaknesses:**

1. As compared with existing studies, e.g., MoleculeSTM, the main difference in this paper is to consider motif-level knowledge. Then the similar framework from MoleculeSTM seems can be extended in a straightforward way. Also, motif info is widely used in existing studies (though not in the molecule-text moltimodal scenario). Therefore, the authors need to better explain the novelties of the proposed approach.
2. Modeling both molecule-level and motif-level knowledge may increase computational costs compared to models focusing solely on molecule-level representations, and the computational complexity should be discussed.
3.  The effectiveness of FineMolTex will rely on accurate and meaningful motif extraction from molecular graphs. Can BRICS algorithm used in this paper meet this requirement? or is there other motif extraction algorithms that will show a performance difference?

**Questions:**

Please see the weaknesses.

---

> ### Author Response · Authors · 2024-11-19
> **Rebuttal by Authors**
>
> We sincerely thank you for the precious time spent reading through the paper and giving constructive suggestions. We address your main concerns as follows:
>
> > W1:  As compared with existing studies, e.g., MoleculeSTM, the main difference in this paper is to consider motif-level knowledge. Then the similar framework from MoleculeSTM seems can be extended in a straightforward way. Also, motif info is widely used in existing studies (though not in the molecule-text moltimodal scenario). Therefore, the authors need to better explain the novelties of the proposed approach.
>
> Our main innovation boils down to the exploration of motif-level knowledge in molecular graph-text pre-training. Motifs determine the molecular properties, so it is crucial to incorporate motif-level knowledge during pre-training. We have utilized some existing techniques, but we stress that **no previous solution can be directly applied to learning motif-level knowledge**, because the lack of supervision signals for the alignment between motifs and words. To address the issue, we first fragment molecular graphs and descriptions into tokens of the same granularity within a unified semantic space, and then we design a masked multimodal modeling task, enabling FineMolTex to learn fine-grained alignment knowledge in a self-supervised manner.
>
> > W2: Modeling both molecule-level and motif-level knowledge may increase computational costs compared to models focusing solely on molecule-level representations, and the computational complexity should be discussed.
>
> **As indicated in our paper (line 491), we have provided the comparison of pre-training and inference times are provided in Appendix D**.   As shown in Table 8 in the Appendix, the pre-training and inference times for FineMolTex are comparable to those of the baselines.
>
> As for the computational complexity, while the architecture of FineMolTex, MolCA, and MoleculeSTM are different, they all include a combination of transformer layers and a graph encoder. The total computational complexity can be expressed as  $O(L_1N^2d_1+L_2Ed_2)$, where $d_1$ is the dimension of the input embeddings of the words, $d_2$ is the dimension of the input embeddings of the nodes, $L_2$ is the number of the transformer layers, $L_1$ is the number of the transformer layers, $E$ is the number of the edges of the graph, $N$ is the input sequence length (number of tokens). The actual computational costs differ based on the values of parameters $L_1$, $N$, $d_1$, $L_2$, $E$, $d_2$, which vary across models due to different design choices and input representations. Here we provide the overall parameter of FineMolTex. MolCA, and MoleculeSTM.  We can see that the number of parameters in FineMolTex is significantly smaller than MolCA and is on the same order of magnitude as MoleculeSTM. Meanwhile, our model achieves the best performance across various downstream tasks.
>
> | Model      | FineMolTex | MolCA | MoleculeSTM |
> | ---------- | ---------- | ----- | ----------- |
> | Parameters | 189M       | 1.3B  | 112M        |
>
>
> > W3: The effectiveness of FineMolTex will rely on accurate and meaningful motif extraction from molecular graphs. Can BRICS algorithm used in this paper meet this requirement? or is there other motif extraction algorithms that will show a performance difference?
>
> Most of the motifs extracted by the BRICS algorithm are meaningful; however, some motifs, such as those composed of single atoms, may lack direct chemical significance. Despite this, the masked learning mechanism employed by FineMolTex demonstrates a strong ability to associate meaningful motifs with their corresponding textual descriptions. As shown in Figure 5b of our paper, our model successfully learns fine-grained alignment for the meaningful motifs.
>
> To extract motifs from molecules,  we could fragment molecules into motifs or use RDKit to extract specific motifs within the molecules.  In our work, we utilize BRICS along with a post-processing procedure [1], which has been validated to be effective.  RDKit can also directly identify specific functional groups in molecules. However, such methods are limited to detecting only a predefined set of functional groups (typically dozens) and lack generality. This restricts their application in broader motif-level representation learning. In contrast, our approach aims to generalize to various types of motifs, even those not explicitly predefined, by leveraging the masked multimodal modeling pre-training task.
>
> [1] Zaixi Zhang, Qi Liu, Hao Wang, Chengqiang Lu, and Chee-Kong Lee. Motif-based graph self-supervised learning for molecular property prediction. NeurlPS 2021

---

### Official Review · Reviewer_ZFJD · 2024-11-02

**Soundness:** 2
**Presentation:** 3
**Contribution:** 2
**Rating:** 3
**Confidence:** 5

**Summary:**

The paper presents FineMolTex, a pre-training framework designed to enhance molecular representation learning by integrating coarse-grained molecule-level knowledge with fine-grained motif-level insights.
Recognizing the importance of motifs in tasks that require detailed molecular understanding, FineMolTex employs a two-branch architecture for motif embedding and textual representation learning, incorporating a cross-attention layer to facilitate information exchange between modalities.
The framework utilizes two pre-training tasks: a contrastive alignment task for molecule-level matching and a masked multi-modal modeling task for motif-level matching.
Experimental results indicate performance improvements in downstream tasks, particularly in text-based molecule editing.

**Strengths:**

1. The experimental results demonstrate a promising performance gain over existing baselines.
2. The paper is well-organized, featuring a logical flow and clear explanations that make it easy to follow.

**Weaknesses:**

1. The figures require refinement; in Figures 1 and 2, some highlighted areas extend beyond the dashed boxes, and the <mask> tokens overlap with the text and motifs (e.g., in Figure 2(b), did you input "carboxylic"?).
2. Retrieve tasks need more metrics like recall.
3. The experiments and datasets generally follow the MoleculeSTM framework but overlook more challenging and practical text-based molecule editing tasks, such as multiple-objective property-based editing, binding-affinity-based editing, and drug relevance editing. This omission significantly undermines the claim of "a notable improvement of up to 230% in the text-based molecule editing task."

**Questions:**

I fully acknowledge the significance of motif-level molecule-text alignment, which the authors assert as their primary contribution.
However, I did not find any explicit supervision signal for such fine-grained alignment.
I remain unconvinced that masked multi-modal modeling can effectively capture it.
Furthermore, the experiments lack qualitative results that would demonstrate the effectiveness of this alignment, aside from the case studies.
Given that this is the most critical claim made by the authors, I would reconsider my rating to accept if I am convinced that masked multi-modal modeling successfully achieves motif-level molecule-text alignment.

---

> ### Author Response · Authors · 2024-11-21
> **Rebuttal by Authors (1/2)**
>
> Thanks sincerely for your careful reading and thought-provoking insights. We address your main concerns as follows:
>
> > W1: The figures require refinement; in Figures 1 and 2, some highlighted areas extend beyond the dashed boxes, and the <mask> tokens overlap with the text and motifs (e.g., in Figure 2(b), did you input "carboxylic"?).
>
> Thank you for your suggestion. The input word is indeed "carboxylic." We have carefully refined the figure. The updated figure will be included in the revised version of the paper.
>
> > W2: Retrieve tasks need more metrics like recall.
>
> Following the testing procedure of [1], we also utilize accuracy as the evaluation metric, as we believe it is sufficient to compare the performance of the models in our tasks. As you suggested, we have included the recall@20 metric in our evaluation. The recall@20 performance on the molecule-ATC, DrugBank-Description, and DrugBank-Pharmacodynamics datasets are presented in the tables below. As shown in the tables below, FineMolTex consistently outperforms baselines.
>
> |             | Given Molecular Graph | Given Text |
> | ----------- | --------------------- | ---------- |
> | MoleculeSTM | 18.9                  | 16.3       |
> | MolCA       | 21.0                  | 22.1       |
> | FineMolTex  | $\textbf{24.1}$                  | $\textbf{23.0}$       |
>
> molecule-ATC
>
> |             | Given Molecular Graph | Given Text |
> | ----------- | --------------------- | ---------- |
> | MoleculeSTM | 86.0                  | 78.8       |
> | MolCA       | 72.5                  | 79.2       |
> | FineMolTex  | $\textbf{88.3}$                  | $\textbf{87.4}$       |
>
> DrugBank-Description
>
> |             | Given Molecular Graph | Given Text |
> | ----------- | --------------------- | ---------- |
> | MoleculeSTM | 60.8                  | 56.3       |
> | MolCA       | 59.7                  | 62.7       |
> | FineMolTex  | $\textbf{73.0}$                  | $\textbf{78.2}$       |
>
> DrugBank-Pharmacodynamics
>
> > W3: The experiments and datasets generally follow the MoleculeSTM framework but overlook more challenging and practical text-based molecule editing tasks, such as multiple-objective property-based editing, binding-affinity-based editing, and drug relevance editing. This omission significantly undermines the claim of "a notable improvement of up to 230% in the text-based molecule editing task.
>
> - Multiple-objective editing:
>
> We would like to clarify that **we have conducted the multiple-objective property-based editing** experiments, as shown in the second line of Figure 3 in the paper. These experiments demonstrate FineMolTex's capability to handle multiple objectives simultaneously, showcasing its effectiveness in achieving complex property-based editing tasks.
>
> - Drug-relevance editing:
>
>  As you suggected, we have conducted experiments on drug relevance editing, and the results are summarized in the table below. As observed, FineMolTex consistently outperforms other baselines, highlighting its superior performance in this task.
>
> |             | This molecule looks like aspirin | This molecule looks like Caffeine. |
> | ----------- | -------------------------------- | ---------------------------------- |
> | MoleculeSTM | 52.0                             | 50.5                               |
> | MolCA       | 48.0                             | 44.5                               |
> | FineMolTex  | 66.5                             | 59.0                               |
>
> - Binding-affinity-based editing:
>
>  The evaluation of the binding-affinity-based editing task conducted by [1] relies on pre-trained evaluation model. Unfortunately, this code has not yet been provided. To ensure accurate and fair comparisons, we are currently in the process of obtaining the code and training details for the evaluation model. Once the code and training details are available, we will include our results on the binding-affinity-based editing task in the revised version of the paper.

---

> ### Author Response · Authors · 2024-11-21
> **Rebuttal by Authors (2/2)**
>
> Q: Qualitative results that demonstrate the effectiveness of fine-grained alignment
>
> **Supervision Signal for Fine-Grained Alignment:** To the best of our knowledge, there are no supervised signals for the motif-text alignment. So we use the masked multi-modal modeling (MMM) task to provide an implicit but effective supervision signal for fine-grained motif-text alignment. Specifically, during pre-training, motifs and text tokens are masked and their labels are predicted using tokens from both modalities. This design, coupled with the cross-attention mechanism, allows the model to align motifs and corresponding text descriptions. For instance, as shown in Figure 2 of the paper, the masked motif (e.g., SO₃⁻) can be reconstructed based on the relevant textual tokens such as "propanesulfonic," thus embedding motif-text relationships into the learned representations.
>
> **Qualitative Demonstration in Text-Based Molecule Editing Task:**
> The results of the text-based molecule editing task provide strong evidence that FineMolTex has effectively learned motif-level knowledge. In this task, the model is instructed to generate molecules with specific motifs or properties. This task requires the model to recognize motif names and understand their properties. And FineMolTex significantly outperforms all baselines. Furthermore, we have included **visual qualitative examples** (Figure 4 in the paper and Figure 8 in the Appendix) that illustrate how the model replaces specific motifs or modifies molecules to exhibit desired motifs or properties. These results qualitatively confirm the model's ability to capture and utilize motif-level knowledge.
>
> **Qualitative Demonstration with LIME Interpretability:** Our experiment, LIME-based interpretability analyses (Figure 5b and Figure 9 in the Appendix), have provided qualitative results that demonstrate the effectiveness of the alignment of our model. Specifically, we use our model to predict masked motifs based on text tokens. We can see that text tokens which are relevant to the masked motif indeed have more contribution to the predicting task.
>
> We would greatly appreciate it if you could specify the type of qualitative demonstration that you would find most convincing. We are more than willing to provide additional results or experiments to meet your expectations and further enhance the presentation of our work.
>
> Thanks again for your careful review. We will diligently revise the presentation of our paper and ensure that all experimental results discussed in this rebuttal are incorporated into the revised version.
>
> [1] Liu S, et al. Multi-modal molecule structure–text model for text-based retrieval and editing[J]. Nature Machine Intelligence 2023

---

### Official Review · Reviewer_eDwk · 2024-11-04

**Soundness:** 2
**Presentation:** 3
**Contribution:** 2
**Rating:** 3
**Confidence:** 4

**Summary:**

The paper proposes FineMolTex, a framework designed for fine-grained molecular graph-text pre-training, focusing on motif-level knowledge to bridge molecular graphs and textual descriptions. The paper claims novelty in learning fine-grained motif knowledge alongside coarse molecule-level knowledge. FineMolTex employs two key tasks: contrastive alignment for molecule-text matching and masked multi-modal modeling for motif-level alignment.

**Strengths:**

- FineMolTex emphasizes motif-level knowledge. This attention to motifs could improve understanding of molecular properties crucial for zero-shot tasks.
- The use of contrastive alignment and masked multi-modal modeling helps integrate fine-grained motif and molecule-level knowledge.
- FineMolTex demonstrates improved performance on tasks like graph-text retrieval and molecule editing.

**Weaknesses:**

- The paper claims, “We are the first to reveal that learning fine-grained motif-level knowledge provides key insight for bridging molecular graphs and text descriptions.” However, prior work, such as HIGHT [1], has already established the importance of motif-level knowledge for improving alignment and preventing hallucination. HIGHT also introduces a hierarchical graph tokenizer that captures information at the node, motif, and graph levels. The motivation and core idea behind your paper and HIGHT are essentially the same.
- The core architecture of FineMolTex lacks novelty. The contrastive pretraining and cross-attention mechanisms for different modalities are derived from BLIP-2, while the masked modeling approach is taken from BERT and is commonly used in models like MAE. The methodology does not present any surprising innovations.
- The architecture of FineMolTex appears no more advanced than Q-Former in BLIP-2 and lacks key pretraining tasks such as Image-Text Matching and Image-Grounded Text Generation present in Q-Former. It is unclear why the authors propose an architecture seemingly weaker than Q-Former instead of directly leveraging Q-Former itself.
- The experiments could be expanded to include more tasks, such as molecule captioning and generation.
- The paper compares against older baselines and omits recent baselines like 3D-MoLM [2]. This casts doubt on the reported state-of-the-art performance.
- Key hyperparameter details are not provided.

[1] Chen, Y., Yao, Q., Zhang, J., Cheng, J., & Bian, Y. (2024). HIGHT: Hierarchical Graph Tokenization for Graph-Language Alignment. arXiv preprint arXiv:2406.14021.

[2] Li, S., Liu, Z., Luo, Y., Wang, X., He, X., Kawaguchi, K., ... & Tian, Q. (2024). Towards 3d molecule-text interpretation in language models. arXiv preprint arXiv:2401.13923.

**Questions:**

- Do you train the transformer and cross-attention layers from scratch? How many layers are used?
- Why did you choose to train the transformer layers rather than using a pretrained LLM and then finetuning?
- Given that FineMolTex does not use a pretrained LLM, how does its computational efficiency (in terms of training and inference time) and memory cost compare with models that utilize LLMs?

---

> ### Author Response · Authors · 2024-11-22
> **Rebuttal by Authors (1/3)**
>
> Thanks sincerely for your careful reading and review. We have addressed your concerns as follows:
>
> > W1: Comparison with HIGHT
>
> We would like to clarify that HIGHT is currently under review for ICLR 2025 and has not yet been formally published. While HIGHT introduces the concept of motif-level knowledge, it detects **only 38 predefined motifs** for molecules and pre-train the model to learn **only names** of these motifs. This limitation restricts its generality and applicability in broader motif-level representation learning. In contrast, our approach focuses on an **open vocabulary**. By fragmenting molecules into motifs and texts into words, and employing the masked multi-modal modeling task, FineMolTex learns fine-grained alignment in a self-supervised manner. This design enables FineMolTex to generalize to a diverse range of motifs, including those not explicitly predefined, and to acquire comprehensive knowledge about motifs, including their names and properties. For instance, as demonstrated in Figure 5(b) of our paper:
>
> - In Figure 5(b)(1), FineMolTex successfully learns the name of the masked token "triazole."
> - In Figure 5(b)(2), it captures the aromatic property of the benzene ring.
> - In Figure 5(b)(3), it not only identifies the name of the complex motif "organonitrogen" but also learns that this motif is "heterocyclic."
>
> This ability to learn both the names and properties for diverse motifs demonstrates the broader scope and flexibility of FineMolTex.
>
> Additionally, while HIGHT includes node-level information, we would like to emphasize that the information contained within a single node is rather limited [2]. In molecular graphs, critical insights are better represented at the motif or graph level, which is the focus of our work.
>
> > W2: Concern about the architecture
>
> Our main innovation boils down to the exploration of motif-level knowledge in molecular graph-text pre-training. Motifs determine the molecular properties, so it is crucial to incorporate motif-level knowledge during pre-training. We have utilized some existing techniques, but we stress that **no previous solution can be directly applied to learning motif-level knowledge**, because the lack of supervision signals for the alignment between motifs and words. To address the issue, we first fragment molecular graphs and descriptions into tokens of the same granularity within a unified semantic space, and then we design a masked multimodal modeling task, enabling FineMolTex to learn fine-grained alignment knowledge in a self-supervised manner.
>
> > W3: Concern about not utilizing the architecture of BLIP-2
>
> We would like to clarify that **the architecture of BLIP-2 cannot be directly utilized for learning fine-grained alignment in our context**. BLIP-2 relies on **explicit supervision signals** to establish alignment relationships, such as Image-Text Matching and Image-Grounded Text Generation. However, we lack explicit supervision signals for motif-level alignment. To address this, FineMolTex fragments molecules into motifs and texts into words and employs the masked multi-modal modeling task. This approach enables FineMolTex to learn a wide variety of motif-level knowledge in a self-supervised manner, capturing both the names and properties of motifs without requiring explicit supervision.
>
> Additionally, our experimental results demonstrate that FineMolTex significantly outperforms MolCA [2], a model based on the BLIP-2 architecture, across multiple tasks. These results highlight the effectiveness of FineMolTex for learning fine-grained representation learning.
>
> > W4: The experiments could be expanded to include more tasks, such as molecule captioning and generation.
>
> Following the testing procedure of [1], we have already evaluated the performance of our model on three downstream tasks: graph-text retrieval, molecule property prediction, and text-based molecule editing. Notably, **text-based molecule editing is a form of molecule generation**, which we believe is practical in real-world applications.
>
> The experimental results across these tasks demonstrate the effectiveness of our model in capturing fine-grained knowledge. We appreciate your suggestion and will consider expanding our experiments to include additional tasks in future work.

---

> ### Author Response · Authors · 2024-11-22
> **Rebuttal by Authors (2/3)**
>
> > W5: Comparison to 3D-MoLM
>
> Thank you for your suggestion. We have conducted experiments on the graph-text retrieval task on DrugBank-Pharmacodynamics, molecule-ATC, and DrugBank-Description. The experimental results for 3D-MoLM and other models are provided in the tables below. As shown in the tables, FineMolTex consistently outperforms 3D-MoLM, demonstrating its effectiveness in capturing fine-grained knowledge. We will carefully include these results in the revised version of our paper.
>
> |             |                  | **Given Molecular Graph** |                  |                  | **Given Text**   |                  |
> | ----------- | ---------------- | ------------------------- | ---------------- | ---------------- | ---------------- | ---------------- |
> | **T**       | **4**            | **10**                    | **20**           | **4**            | **10**           | **20**           |
> | KV-PLM      | 68.38 ± 0.03     | 47.59 ± 0.03              | 36.54 ± 0.03     | 67.68 ± 0.03     | 48.00 ± 0.02     | 34.66 ± 0.02     |
> | MolCA       | 83.75 ± 0.54     | 74.25 ± 0.26              | 66.14 ± 0.21     | 81.27 ± 0.33     | 69.46 ± 0.17     | 62.13 ± 0.16     |
> | MoMu-S      | 70.51 ± 0.04     | 55.20 ± 0.15              | 43.78 ± 0.10     | 70.71 ± 0.22     | 54.70 ± 0.31     | 44.25 ± 0.43     |
> | MoMu-K      | 69.40 ± 0.11     | 53.14 ± 0.26              | 42.32 ± 0.28     | 68.71 ± 0.03     | 53.29 ± 0.05     | 43.83 ± 0.12     |
> | MoleculeSTM | 92.14 ± 0.02     | 86.27 ± 0.02              | 81.08 ± 0.05     | 91.44 ± 0.02     | 86.76 ± 0.03     | 81.68 ± 0.03     |
> | 3D-MoLM     | 81.35 ± 0.14     | 73.65 ± 0.13              | 64.79  ± 0.15    | 79.78±0.22       | 62.38  ± 0.16    | 53.43±0.11       |
> | FineMolTex  | **95.86 ± 0.34** | **91.95 ± 0.06**          | **85.80 ± 0.05** | **95.80 ± 0.06** | **92.18 ± 0.12** | **85.01 ± 0.32** |
>
> DrugBank-Pharmacodynamics
>
> |             |                  | **Given Molecular Graph** |                  |                  | **Given Text**   |                  |
> | ----------- | ---------------- | ------------------------- | ---------------- | ---------------- | ---------------- | ---------------- |
> | **T**       | **4**            | **10**                    | **20**           | **4**            | **10**           | **20**           |
> | KV-PLM      | 60.94 ± 0.00     | 42.35 ± 0.00              | 30.32 ± 0.00     | 60.67 ± 0.00     | 40.19 ± 0.00     | 29.02 ± 0.00     |
> | MolCA       | 67.34 ± 0.05     | 53.51 ± 0.12              | 44.10 ± 0.03     | 65.18 ± 0.34     | 51.01 ± 0.26     | 41.30 ± 0.51     |
> | MoMu-S      | 64.72 ± 0.04     | 48.72 ± 0.03              | 37.64 ± 0.02     | 64.98 ± 0.13     | 49.58 ± 0.05     | 39.04 ± 0.16     |
> | MoMu-K      | 61.79 ± 0.14     | 45.69 ± 0.22              | 34.55 ± 0.09     | 63.32 ± 0.15     | 47.55 ± 0.06     | 37.68 ± 0.18     |
> | MoleculeSTM | 69.33 ± 0.03     | 54.83 ± 0.04              | 44.13 ± 0.05     | 71.81 ± 0.05     | 58.34 ± 0.07     | 47.58 ± 0.05     |
> | 3D-MoLM     | 65.72 ± 0.08     | 50.48 ± 0.14              | 38.31 ± 0.06     | 63.10  ± 0.06    | 44.17  ± 0.11    | 34.56  ± 0.15    |
> | FineMolTex  | **75.43 ± 0.15** | **60.66 ± 0.08**          | **49.45 ± 0.24** | **75.22 ± 0.12** | **60.29 ± 0.04** | **48.42 ± 0.15** |
>
> molecule-ATC
>
> |             | **Given Molecular Graph** |                  |                  | **Given Text**   |                  |                  |
> | ----------- | ------------------------- | ---------------- | ---------------- | ---------------- | ---------------- | ---------------- |
> | **T**       | **4**                     | **10**           | **20**           | **4**            | **10**           | **20**           |
> | KV-PLM      | 73.80 ± 0.00              | 53.96 ± 0.29     | 40.07 ± 0.38     | 72.86 ± 0.00     | 52.55 ± 0.29     | 40.33 ± 0.00     |
> | MolCA       | 93.75 ± 0.09              | 87.25 ± 0.06     | 82.77 ± 0.12     | 90.71 ± 0.04     | 84.97 ± 0.16     | 77.53 ± 0.15     |
> | MoMu-S      | 76.52 ± 0.12              | 61.66 ± 0.25     | 50.00 ± 0.08     | 77.62 ± 0.06     | 61.49 ± 0.15     | 52.20 ± 0.13     |
> | MoMu-K      | 74.15 ± 0.08              | 57.18 ± 0.16     | 47.97 ± 0.14     | 77.79 ± 0.12     | 62.33 ± 0.18     | 47.97 ± 0.06     |
> | MoleculeSTM | 99.15 ± 0.00              | 97.19 ± 0.00     | 95.66 ± 0.00     | 99.05 ± 0.37     | 97.50 ± 0.46     | 95.71 ± 0.46     |
> | 3D-MoLM     | 92.81±0.23                | 85.71± 0.19      | 80.20± 0.33      | 88.31±0.32       | 81.23± 0.07      | 74.40±0.39       |
> | FineMolTex  | **99.58 ± 0.05**          | **97.97 ± 0.00** | **96.45 ± 0.16** | **99.58 ± 0.04** | **97.89 ± 0.08** | **96.11 ± 0.12** |
>
> DrugBank-Description

---

> ### Author Response · Authors · 2024-11-22
> **Rebuttal by Authors (3/3)**
>
> > W6: Key hyperparameter details are not provided.
>
> **As indicated in our paper (line 265), details of the pre-training process can be found in Appendix C.4**, where we provide the key hyperparameter details. We will revise the corresponding parts in the paper to ensure these details are more explicitly referenced and easier to locate.
>
> > Q1: Do you train the transformer and cross-attention layers from scratch? How many layers are used?
>
> The transformer and cross-attention layers in our model are trained from scratch. The number of cross-attention layers is set to 2, resulting in 2 rounds of cross-modal interaction. For the molecule stream, the transformer layers are set to 2 in the first round and 3 in the second round. For the text stream, the transformer layers are set to 10 in the first round and 12 in the second round.
>
> > Q2: Why did you choose to train the transformer layers rather than using a pretrained LLM and then finetuning?
>
> In our model, the transformer layers are not used to encode texts directly but to transform molecular graphs and texts into a joint embedding space and to learn fine-grained alignment representations. This specific purpose requires training these layers from scratch, as a pre-trained LLM cannot effectively perform this task.
>
> We do utilize a pre-trained text encoder, SciBERT, to encode the text input. Our framework is flexible, and we could replace it with a pre-trained LLM.
>
> > Q3: Given that FineMolTex does not use a pretrained LLM, how does its computational efficiency (in terms of training and inference time) and memory cost compare with models that utilize LLMs?
>
> **As indicated in our paper (line 491) the comparison of pre-training and inference times are provided in Appendix D**. As shown in Table 8 in our Appendix,  the pre-training and inference times for FineMolTex are comparable to those of the baselines.
>
> We provide the overall parameter of FineMolTex. MolCA, and MoleculeSTM. We can see that the number of parameters in FineMolTex is significantly smaller than MolCA and is on the same order of magnitude as MoleculeSTM. Meanwhile, our model achieves the best performance across various downstream tasks.
>
> | Model      | FineMolTex | MolCA | MoleculeSTM |
> | ---------- | ---------- | ----- | ----------- |
> | Parameters | 189M       | 1.3B  | 112M        |
>
> [1] Liu S, et al. Multi-modal molecule structure–text model for text-based retrieval and editing[J]. Nature Machine Intelligence 2023
>
> [2] Liu Z, et al. MolCA: Molecular graph-language modeling with cross-modal projector and uni-modal adapter. EMNLP 2023

---

### Official Review · Reviewer_dJDd · 2024-11-09

**Soundness:** 3
**Presentation:** 2
**Contribution:** 2
**Rating:** 5
**Confidence:** 4

**Summary:**

This paper introduces FineMolTex, a multi-modal learning framework for molecule-text modeling. FineMolTex is a multi-modal language model that jointly models molecules and texts. For molecules, it decomposes them into motifs, and utilize a GNN to obtain motif embedding; for text, it utilizes a pretrained BERT encoder to obtain text embeddings. Then, the motif embeddings and text embeddings are fed into separate transformers for cross-modal learning. Specifically, FineMolText utilizes two pretraining tasks: 1) contrastive alignment and 2) masked multi-modal modeling. For contrastive alignment, a classifical contrastive learning loss is applied on the final embedding of motifs and texts, obtained from separate transformers. For masked multi-modal modeling, motifs and texts are randomly masked, and the transformers are trained to recover the masked tokens using Cross-Entropy loss. Notably, for this task, the two transformers for texts and motifs are connected through the internal cross-attention layers.

The proposed method are further applied for downstream tasks of graph-text retrieval (Table 1, Table 2), molecular property prediction (Table 3), and molecule editing (Figure 3, Figure 4).

**Strengths:**

1. The proposed method is overall sound, and the studied problem is relevant to the ICLR conference.
2. The proposed method achieves top performances for graph-text retrieval for the DrugBank-Pharmacodanamics, and molecule-ATC datasets.

**Weaknesses:**

1. The overall methodology is not surprising. Most components, like the multi-modal masked modeling and contrastive learning, are already seen in previous works. The new part is to represent molecules as decomposed motifs and use GNN encoder for motif representation.
2. Considering PubChem is used as as the training dataset, you need to test your model on PubChem's test set to really demonstrate the performance of your model. This is standard in your baselines, like MoleculeSTM, MoMu, and MolCA.
3. The authors have tested their model for property prediction on the MoleculeNet datasets. However, as I understand, the value of this evaluation is insignificant. The main reason is the limited performances for the proposed method and all the baselines. As shown in [1], combining proper feature engineering and simple algos, like SVM, usually achieve much better performances than deep learning models. Therefore, the authors should explain the value of this evaluation.

**Reference:**

[1] Understanding the Limitations of Deep Models for Molecular property prediction: Insights and Solutions. In NeurIPS 2023.

**Questions:**

1. The motivation of this work is to study fine-grained molecule representation (motifs). Have the authors considered combining a global representation, like a global GNN embedding and a complete SMILES, with fine-grained representations for improved performance?
2. Does using fine-grained representation of motifs improve the explanability of your method?

---

> ### Author Response · Authors · 2024-11-18
> **Rebuttal by Authors**
>
> Thanks sincerely for your careful reading and thought-provoking insights. We address your main concerns as follows:
>
> > W1: Concern about innovation.
>
> Our main innovation boils down to the exploration of motif-level knowledge in molecular graph-text pre-training. Motifs determine the molecular properties, so it is crucial to incorporate motif-level knowledge during pre-training. We have utilized some existing techniques, but we stress that **no previous solution can be directly applied to learning motif-level knowledge**because the lack of supervision signals for the alignment between motifs and words. To address the issue, we first fragment molecular graphs and descriptions into tokens of the same granularity within a unified semantic space, and then we design a masked multimodal modeling task, enabling FineMolTex to learn fine-grained alignment knowledge in a self-supervised manner.
>
> > W2:  Concern about testing on PubChem dataset
>
> **Indeed, MoleculeSTM [2] was not tested on the PubChem dataset.** Instead, [2] evaluated the model's performance on the DrugBank database, which provides valuable opportunities for exploring drug discovery tasks. And we follow their testing procedure. Specifically, three fields from DrugBank were utilized: the Description field, the Pharmacodynamics field, and the Anatomical Therapeutic Chemical (ATC) field. We believe that assessing model performance on datasets directly relevant to drug discovery offers more meaningful insights than testing solely on the PubChem dataset.
>
> > W3: Concern about testing on property prediction task.
>
> Previous multimodal molecular graph learning methods [2, 3, 4] have consistently evaluated their models on molecular property prediction tasks. Following their work, we include this evaluation to ensure comparability and to demonstrate that FineMolTex learns better representations for molecules.
>
> > Q1: Combining a global representation with fine-grained representations.
>
> Indeed, we use the same graph encoder, text encoder, and transformer layer for learn fine-grained representation and global representation. Thus the global representation inherently captures fine-grained knowledge as well. As you suggested, we test whether combining molecule representations with motif representations would improve performance. The results on the molecule-ATC dataset are shown in the table below. As observed, this combination leads to performance degradation. This may be because some motifs, such as single atoms, have little intrinsic meaning. These motifs are better suited to being part of the molecule representation, as considering them separately can introduce redundant information.
>
> |                                                         |                 | Given Molecular Graph |                 |                 | Given Text     |                |
> | ------------------------------------------------------- | --------------- | --------------------- | --------------- | --------------- | -------------- | -------------- |
> | Model                                                   | T=4             | T=10                  | T=20            | T=4             | T=10           | T=20           |
> | FineMolTex                                              | **75.43±0.15**  | **60.66±0.08**        | **49.45±0.24**  | **75.22±0.12**  | **60.29±0.04** | **48.42±0.15** |
> | FineMolTex (combine molecule and motif representations) | $73.25\pm 0.12$ | $57.34\pm 0.06$       | $44.34\pm 0.13$ | $72.24\pm 0.10$ | $56.37\pm0.08$ | $44.80\pm0.09$ |
>
>
> > Q2: Does using fine-grained representation of motifs improve the explanability of your method?
>
> Yes, using fine-grained representations of motifs improves the explainability of our method. In Section 4.4, "Predictions Based on Fine-Grained Knowledge," we predict the labels of masked motifs using text tokens and employ LIME [5] to quantify the significance of each text token in predicting these motifs. As shown in Figure 5b of our paper, text tokens with higher interpretive weights play a crucial role in the predictions. Importantly, we observe that these tokens are indeed relevant to the masked motifs, demonstrating the enhanced explainability provided by the fine-grained representations.
>
> [1] Understanding the Limitations of Deep Models for Molecular property prediction: Insights and Solutions. In NeurIPS 2023.
>
> [2] Liu S, et al. Multi-modal molecule structure–text model for text-based retrieval and editing[J]. Nature Machine Intelligence 2023
>
> [3] Liu Z, et al. MolCA: Molecular graph-language modeling with cross-modal projector and uni-modal adapter. EMNLP 2023
>
> [4] Su B, et al. A molecular multimodal foundation model associating molecule graphs with natural language, arXiv 2022
>
> [5] Ribeiro M T, et al. " Why should I trust you?" Explaining the predictions of any classifier. KDD 2016

---

> > ### Comment · Reviewer_dJDd · 2024-11-24
> > **Further questions**
> >
> > Q1. About the new experiment of `Combining a global representation with fine-grained representations`, how is this experiment conducted? Specifically, did you use the same fingerprint feature as the [1]? Did you retrain your model using these fingerprints, or did you use your original checkpoint but incorporating these features by concatenation? Knowing these details is important to understand the true effect of incorporating molecular fingerprint.
> >
> >
> >
> > Reference:
> > [1] Understanding the Limitations of Deep Models for Molecular property prediction: Insights and Solutions. In NeurIPS 2023.

---

> > > ### Author Response · Authors · 2024-11-25
> > > **Response to Q1**
> > >
> > > Thank you for your question!
> > >
> > > We did not use the fingerprint feature as described in [1]. Instead, since our model can already generate molecular embeddings and motif embeddings, we directly utilized these embeddings from our original checkpoint. Specifically, we obtained two types of embeddings: the molecular embedding and the motif embeddings for each molecule. Because different molecules contain varying numbers of motifs, we calculated the average of all motif embeddings for each molecule to derive a single motif representation. While using a weighted average based on motif importance would likely yield better results, it would require additional training to determine this importance, so we opted for the averaging approach for simplicity. Finally, we concatenated the averaged motif embedding with the molecular embedding to form the combined representation.
> > >
> > > For the Graph-Text retrieval task, we calculated the similarity between the molecular embedding and the text embedding to identify the corresponding pairs. As the dimensionality of the molecular embedding doubled due to concatenation, we also concatenated the text embedding with itself to ensure dimensional consistency.
> > >
> > > We hope this explanation clarifies our methodology. Please feel free to reach out if you have further questions or need additional details.
> > >
> > > [1] Understanding the Limitations of Deep Models for Molecular property prediction: Insights and Solutions. In NeurIPS 2023.

---

### Note · Authors · 2025-01-14

I have read and agree with the venue's withdrawal policy on behalf of myself and my co-authors.